# The Interplay Between Psychological and Neurobiological Predictors of Weight Regain: A Narrative Review

**DOI:** 10.3390/nu17101662

**Published:** 2025-05-13

**Authors:** Małgorzata Moszak, Justyna Marcickiewicz, Marta Pelczyńska, Paweł Bogdański

**Affiliations:** 1Department of Obesity and Metabolic Disorder Treatment and Clinical Dietetics, Poznań University of Medical Sciences, 49 Przybyszewskiego Street, 60-355 Poznan, Poland; mpelczynska@ump.edu.pl (M.P.); pbogdanski@ump.edu.pl (P.B.); 2Faculty of Medicine, Poznań University of Medical Sciences, 70 Bukowska Street, 60-812 Poznan, Poland; just.marcickiewicz@gmail.com

**Keywords:** obesity, weight control, reward-related eating, hedonic hunger, personality, psychological traits, brain function, motivation, self-control, eating behavior

## Abstract

**Introduction**: Obesity is a global health problem requiring effective interventions to achieve weight loss and maintain it in the long term. A major challenge for clinicians is weight regain (WR), defined as progressive weight gain following successful weight loss. WR is affected by multiple factors, including psychological traits linked to specific brain alterations. Understanding these mechanisms is crucial in developing strategies to prevent WR and to ensure effective weight control. **Objectives**: This narrative review aims to gather current findings on the psychological and neurobiological determinants of WR and to discuss the interplay between these factors. **Methods**: A literature search was conducted on PubMed, Medline, and Web of Science for English-language studies published between December 1990 and November 2024. **Results**: WR is driven by interconnected psychological and neurobiological factors that influence eating behavior and the regulation of body weight. Certain personality traits and emotional patterns are associated with specific changes in brain activity, which together affect vulnerability to WR. Although distinct mechanisms can be identified, the complexity of homeostatic and nonhomeostatic appetite control suggests that no single factor predominates. **Conclusions**: This review highlights the dynamic interplay between psychological and neurobiological predictors of WR. However, due to the narrative nature of this review, the focus on selected determinants, and the limited quality and size of the available studies, further research is needed to comprehensively understand causality and to improve relapse prevention strategies.

## 1. Introduction

Obesity is characterized by excessive fat accumulation leading to negative psychological health outcomes and increasing the risk of several chronic diseases, including diabetes, heart disease, fatty liver disease, infertility, and some types of cancer [1]. It is a global health problem that affects people of all ages, genders, and ethnicities [2]. In 2021, the number of adults aged over 24 years affected by overweight or obesity was estimated to be around 2.11 billion worldwide, almost half of the total adult population, while the number of individuals with overweight or obesity in the pediatric population was 493 million worldwide [3,4].

Greater research attention is thus required on effective interventions for obesity that ensure effective weight loss and successful long-term weight loss maintenance. The first-line therapy for obesity is lifestyle change, including diet, physical activity, and behavioral changes. The holistic approach to obesity treatment also includes pharmacological therapy, bariatric surgery (BS), and psychological support [5,6,7,8]. Due to the close relationship between psychological factors—including behavioral, cognitive, and emotional processes—and weight control, psychological support also plays an important role in the holistic care of obesity [9,10].

A key challenge in obesity management is weight regain (WR), defined as progressive weight gain that occurs after successful treatment-induced weight loss [11]. WR is most common in patients who have lost weight by following lifestyle modifications, regardless of the dietary or behavioral intervention used. According to available data, about 30–35% of the reduced weight is regained within one year after the behavioral intervention, and half of patients return to their initial weight five years after successful weight loss [12]. A range of factors contribute to WR, including genetic predisposition (gene polymorphism) [13] and food environment. Critical factors in long-term weight control failure include biological mechanisms, including metabolic adaptation, endocrine dysfunction, and adipocyte inflammation [11,14]. Additionally, psychological factors, including stress, emotional eating, and individual personal traits, in combination with physiological and environmental factors can have an effect on WR (Figure 1) [15,16]. A full description and understanding of the mechanisms related to WR is necessary to discover interventions that attenuate these mechanisms and, consequently, guarantee effective, long-term weight control in patients with obesity. Many previous studies have explored this topic; however, they have often lacked a comprehensive analysis that integrates both psychological and neurobiological factors. These factors are critical for understanding the full spectrum of the WR phenomenon but are often addressed separately in the literature.

This paper therefore aims to gather current findings on selected determinants of WR—both psychological (e.g., personality traits, emotional regulation strategies, and dichotomous thinking) and neurobiological (e.g., food addiction, metabolic adaptations, and food cue-related brain activity)—and to discuss the interplay between these factors. Bariatric surgery and pharmacotherapy represent approaches that affect the neuronal tracts involved in appetite motivation, and their impact on weight control and WR is also discussed. Given the broad and interdisciplinary nature of the objectives of the current study, a narrative review was conducted. This type of review enables current knowledge to be integrated and research gaps to be identified, allowing for the synthesis of data and their evaluation using diverse standardized and nonstandardized methods.

## 2. Methods

This narrative review was conducted in accordance with guidelines for narrative reviews (Appendix A) [17]. The literature search was conducted using the electronic databases PubMed, Medline, and Web of Science and focused on peer-reviewed studies published between December 1990 and November 2024.

The following search employing Medical Subject Headings (MeSHs) was used to identify the relevant articles:“Personality Traits” [MeSH] OR “Personality” OR “Emotional regulation” OR “Mental Health” [MeSH] OR “Emotional States” OR “Cognitive Distortions” OR “Dichotomous Thinking”.“Brain” [MeSH] OR “Brain Activity” OR “Neural Activation” OR “Neural Mechanisms” OR “Food Addiction” [MeSH] OR “Dopamine” [MeSH] OR “Reward System” OR “Metabolic Adaptations”.”Weight Gain” [MeSH] OR “Weight Regain” OR “Body Weight” [MeSH] OR “Weight Loss” [MeSH] OR “Weight Maintenance” OR “Weight Control” OR “Obesity” [MeSH] OR “Weight Management”.“Bariatric Surgery” [MeSH] OR “Metabolic Surgery” OR “Pharmacotherapy” [MeSH] OR “Anti-Obesity Agents” [MeSH] OR “Pharmacological Treatment”.(1 AND 3) OR (2 AND 3) OR (3 AND 4).

In addition, a manual search was conducted to investigate relevant references of the publications.

Eligible studies were required to meet the following inclusion criteria: (1) conducted on adult populations (eighteen years or older) with overweight/obesity (BMI ≥ 25 kg/m^2^); (2) published in English; (3) available in full-text format; and (4) addressing the research objective. A variety of article types were taken into account, such as observational and experimental studies, systematic reviews, and meta-analyses. However, studies among the underage population and pregnant women were excluded from this review.

Two reviewers (J.M. and M.M) completed the data search and analysis. The selection of studies was guided by their relevance to the review’s objectives rather than by systematic inclusion criteria. Studies were narratively synthesized, with particular attention paid to the interplay between psychological traits (e.g., conscientiousness, self-efficacy, impulsiveness) and neurobiological mechanisms (e.g., food addiction, metabolic adaptations, and brain activity related to food cues) involved in WR.

The Scale for Assessment of Narrative Review Articles (SANRA) checklist was used to assess the quality of the studies (Appendix A) [18].

## 3. Results

### 3.1. Psychological Determinants of WR

#### 3.1.1. Personality Traits and WR

Psychological factors, such as temperament, character, personality traits, self-efficacy, and the ability to control weight, have been identified as strong predictors of weight regain (WR) [15,16]. Although these findings are largely correlational and not necessarily causal, they underscore the important role of psychological profiles in weight management. Personality traits, which are shaped by both genetic and environmental factors, affect lifestyle habits, including eating behaviors. Specifically, reduced self-efficacy and persistence, coupled with higher levels of novelty-seeking and fearfulness, have been associated with an increased risk of obesity [19,20]. Features such as reward dependence and harm avoidance, which reflect a tendency to seek positive reinforcement while avoiding punishment, also appear to affect body weight and weight control efforts [20,21].

Self-efficacy emerges as a particularly important factor, as it plays a crucial role in both achieving weight loss and maintaining it in the long term. High levels of self-efficacy are associated with persistence in the pursuit of goals, increased physical activity, and more deliberate meal planning, which together help mitigate the risk of excessive caloric intake [22]. Moreover, Buchanan et al. [23] used structural equation modeling (SEM) to demonstrate the relationship between binge eating and decreased self-efficacy. Decreased self-efficacy has been associated with maladaptive eating behaviors, such as binge eating, which can undermine adherence to weight loss interventions and reduce the chances of long-term weight control. It has been proposed that binge eating episodes may lower self-efficacy, thereby creating a self-perpetuating cycle of WR. Low self-efficacy, coupled with inadequate self-control and frequent off-schedule eating episodes, has also been correlated with depressive moods, which may further impair the motivation to modify dietary habits. In turn, unsuccessful attempts at weight loss can adversely affect self-esteem, a factor that is intricately linked to self-efficacy [22,24,25,26]. Beyond self-efficacy, several other personality-related factors have been identified as potential predictors of long-term weight maintenance. Traits such as self-control, food-related self-awareness, motivation, openness to dietary changes, and effective stress management strategies are consistently associated with better outcomes [22,24,25,26,27]. In summary, successful weight maintainers often share a psychological profile characterized by higher conscientiousness, greater emotional stability, and a proactive attitude toward lifestyle changes.

Comparative analysis of successful maintainers and regainers has highlighted notable personality and behavioral differences which influence an individual’s relationship with food and diet engagement. Gold et al. [28] reported that maintainers tended to engage in healthier lifestyle behaviors, and their dietary choices are linked to a higher level of self-control. Conversely, dietary choices among regainers were associated with traditionalism, indicating that individuals who adhere to a specific nutritional model may struggle to comply with dietary principles due to a lower openness to lifestyle changes. Successful maintainers also demonstrated more significant control over food quantity and temptation, as well as better adherence to exercise and sleep routines. Additionally, the maintainers reported significantly higher levels of overall conscientiousness, virtue, order, responsibility, and diligence compared to the regainers [28]. The role of conscientiousness as a protective factor in WR and the association between a low level of conscientiousness and greater fluctuations in body weight throughout an individual’s lifespan are also described in the longitudinal study by Sutin et al. [29], who observed that individuals scoring higher in neuroticism and impulsiveness exhibited greater fluctuations in BMI across adulthood. Traits like agreeableness and impulsiveness were also predictive of greater BMI increases. Notably, impulsiveness and neuroticism have been associated with difficulties in responding effectively to weight loss interventions, most likely due to their association with addictive behaviors that can hinder WR [29,30,31].

The importance of self-regulation skills is also emphasized in intervention studies. Milsom et al. [32], in a long-term follow-up study of women with obesity who completed a lifestyle modification program, found that greater self-control over food intake and stronger meal planning abilities were predictive of successful maintenance of at least 5% weight loss over 3.5 years. These findings underscore the potential role of self-monitoring—an aspect of conscientiousness—as a key strategy for sustaining healthy eating patterns. Overall, intrinsic motivation and self-efficacy consistently emerge as central psychological determinants in preventing WR and supporting the long-term adoption of health-promoting behaviors [22].

#### 3.1.2. Emotional Component

Emotional eating—the tendency to eat in response to negative emotions—plays a crucial role in the development of overweight, obesity, and WR. Recent reports highlight a strong association between depressive symptoms, emotional eating behaviors, and weight gain [33,34]. Notably, the presence of the “sensitivity to reward” trait, which is characterized as a tendency to maintain rewarding activities and to derive pleasure from them [35], is particularly pronounced among individuals with binge eating disorder and co-occurring depressive symptoms [36]. These findings suggest that individuals with greater levels of sensitivity to reward may use food as a compensatory mechanism to cope with distress, especially in stressful situations. Although the pathology of emotional eating remains unclear, previous studies have shown an association between emotional regulation difficulties, eating disorders, and susceptibility to weight control [34,35]. Problems with emotional regulation may undermine the effectiveness of dietary interventions and thus contribute to WR.

Emotional regulation is defined as a set of strategies and processes through which individuals can control how they feel, experience, and express emotions [37]. A strong correlation between chronic stress and weight gain has been demonstrated in individuals with obesity [19,38,39], which may be due to emotion suppression via overeating, as food plays the role of an emotional regulator that modulates mood. The regions of the brain involved in controlling appetitive responses and regulating emotions are closely related and may affect an individual’s food choices in response to various emotional stimuli [40]. Previous research has observed reduced activation of the insula in patients with obesity in response to negative emotions [41]. However, individuals provided persistently increased activation of the right anterior insula when instructed to regulate these emotions. The insula is characterized by functional heterogeneity, as it has been assigned many different properties, including being responsible for processing tastes and emotional experiences via integrating interoceptive signals [42]. These results thus suggest that people with obesity may exhibit impaired emotional regulation via prefrontal brain areas in response to negative stimuli.

Emotional regulation problems may be associated with specific personality traits and brain activity. For instance, the amygdala is recognized as the brain’s emotional center, having effects on mood disorders, emotions, and instincts. Closely linked with activation of the amygdala is neuroticism, in which individuals exhibiting higher neuroticism generally show increased activity of the amygdala [43]. Neuroticism is associated with heightened activation of brain regions such as the amygdala and cingulate cortex, which are involved in anxiety and mood disorders. Highly neurotic individuals tend to experience greater emotional conflict and arousal, which may be due to their tendency to ruminate on internal emotional states. This ruminative tendency leads to a higher degree of internal conflict, potentially causing abnormal amygdala activation [43]. There is also a negative correlation between the connectivity between the amygdala and the anterior cingulate cortex (ACC) and the level of neuroticism [44]. This may suggest that people with higher levels of neuroticism are characterized by weakened control of the ACC over the amygdala. Such connectivity patterns may be associated with a greater vulnerability to developing affective disorders such as depression and anxiety. This means that people with high levels of neuroticism may have difficulties regulating their emotions and controlling their reactions to stressful or negative stimuli. Such emotional regulation problems may in turn predispose highly neurotic individuals to weight cycling via compensatory overeating in response to distressing experiences.

Research on emotional regulation has also shown significant differences associated with gender and age in terms of both behavioral and neurobiological levels. Women are more likely to engage in strategies such as rumination, which may increase their vulnerability to mood disorders, while men are more likely to use strategies to avoid or suppress emotions [45]. Rumination involves repetitive, intrusive thoughts about negative emotions and problems. This strategy reinforces negative mood and appears to impede problem-solving and decision-making. It has been shown to be associated with compulsive behavior, including binge eating, as well as with the development of eating disorders [46]. In the context of distress, this strategy thus may increase the risk of using food as a coping mechanism, particularly among women. The functional neuroimaging (fMRI) study of McRae et al. [47] also found differences between the sexes in terms of brain activity during the use of emotional regulation strategies. These data indicate that men show less engagement of the prefrontal regions and greater suppression of amygdala activity, which suggests the use of more automatic mechanisms of regulation. On the other hand, women show greater activity in areas associated with reward processing which, combined with the tendency to ruminate, emphasizes women’s predisposition to develop food-related mechanisms of regulating emotions. The choice of emotional regulation strategy and flexibility in their use also change over different stages of life: Older people tend to favor adaptive strategies, such as reinterpretation (cognitive reappraisal) and acceptance, which in combination with flexibility and distraction promote better mood regulation [48,49]. Younger adults, due to less flexibility in choosing emotional regulation strategies and their more frequent use of less adaptive methods, may be more susceptible to the development of emotional eating in response to negative experiences.

The inability to recognize and accept emotions can be a serious obstacle in long-term weight control and an indirect cause of WR [50]. Overeating in order to deal with negative emotions leads to a problematic pattern in which eating becomes the only technique available for dealing with difficult experiences. It has been described previously that weight gainers also experience more episodes of binge eating throughout the year [51], which may also be due to the mechanisms involved in achieving changes in body weight over the years. The misidentification of emotional states, the lesser ability to deal with negative stimuli, and the increased susceptibility to stress may lead individuals to seek relief in food and excess calorie intake, making it challenging to prevent WR.

#### 3.1.3. Dichotomous Thinking

Epidemiological data indicate that dietary restraint can be a helpful strategy for dealing with weight gain and successful long-term weight maintenance. However, dietary restraint is typically reduced after weight loss achievement, which drives WR [52]. Dietary restraint occurs in “flexible” and “rigid” forms: The rigid form is accompanied by dichotomous thinking (“all or nothing”, “black and white”), which is a strong predictor WR, as it is a categorical thinking style that can trigger binge eating episodes [53]. Dichotomous thinking is a complex psychological phenomenon that can be considered a cognitive style strongly associated with specific personality traits (such as perfectionism) and “splitting”, with potential relevance to personality pathology [54]. The study of Byrne et al. [55] indicates that dichotomous thinking is the strongest predictor of WR among women who previously lost weight. The study noted that women with higher levels of dichotomous thinking had more difficulty maintaining weight loss within one year of completing a weight loss program. Similar conclusions were drawn in an online survey study assessing the relationship between dichotomous thinking, eating behavior, and WR [52]. Byrne et al. demonstrated that it is the dichotomous beliefs about food and dieting, rather than restrictive eating itself, that serve as the primary predictor of WR. Given that this way of thinking is highly extreme, even a small violation of dietary rules can lead to disinhibition, abandonment of previous assumptions, and compensatory overeating of “forbidden” foods, consequently leading to gradual weight gain. Moreover, dichotomous thinking is also associated with lower weight loss among patients after BS [56]. However, this highly rigid thinking style is a key predictor not only of obesity but more generally of the pathophysiology characteristic eating disorders [57]. Dichotomous thinking may also mediate the relationship between depression and obesity [58]. Individuals with more dichotomous thinking are more likely to abandon any dietary restriction in response to negative stimuli, especially in the presence of depressive symptoms. These findings highlight the association between this very extreme thinking style and the increased likelihood of weight cycling throughout the lifespan, particularly as a result of emotional eating.

Flexible cognitive restraint, in contrast, is a weight management approach that allows for a limited quantity of high-calorie foods, rather than totally excluding it. This flexible approach is associated with a greater likelihood of maintaining weight loss than the high level of rigid cognitive constraint found in the dichotomous approach—in which periods of calorie restriction alternate with periods of excessive consumption of calorie-rich foods [59]. Sairanen et al. [60] investigated the effects of both the rigid and flexible approaches on weight management and psychological well-being among 49 overweight participants during a weight loss intervention and in the eight to nine months of the follow-up period. They observed that greater improvements in flexible cognitive restraint during the weight loss intervention were associated with more successful weight maintenance and enhanced well-being. Increases in flexible restraint throughout the intervention correlated with greater reductions in psychological distress. Additionally, a greater decrease in rigid restraint between the intervention and the follow-up assessment was linked to better preservation of psychological well-being at the end of the follow-up period [60].

A flexible eating style and a flexible way of thinking about food seem themselves to be beneficial for weight management. However, restrictive diets that shape inappropriate relationships with food can be either a trigger or a result of dichotomous thinking, which is associated with poorer weight control. Cristea et al. [61] conducted a meta-analysis of 26 randomized clinical trials among depressed patients to assess the effect of cognitive behavioral therapy (CBT) on dysfunctional thinking. The results showed a moderate effect of CBT in reducing dysfunctional thinking compared with control groups, but these effects were strongly correlated with improvements in depressive symptoms. Similarly, Ezawa and Hollon [62] analyzed four studies with a total of 353 patients to assess the effectiveness of cognitive restructuring (CR) as part of psychotherapy. The meta-analysis showed a significant association between the use of CR and improved treatment outcomes, suggesting that CR is an effective technique for modifying maladaptive thinking patterns. The perception of a reduction diet in terms of black-and-white thinking, with food categorized as either “good” or “bad”, seems to undermine motivation for change and can lead to binge eating that appears as an inherent part of dieting. Previous results suggest that there is an association between dichotomous thinking, restraint eating, and WR, as highly rigid thinking during weight loss interventions may decrease appetite control and increase food cravings, leading to overconsumption of forbidden foods. Dichotomous thinking, combined with the lack of proper support during diet therapy, can negatively affect long-term weight control. Given these findings, it is crucial to implement therapeutic interventions, such as CBT or cognitive restructuring, for individuals with dichotomous thinking. These approaches could help address dysfunctional eating patterns, reduce the risk of binge eating, and ultimately prevent WR, supporting more sustainable weight management in the long term.

These findings suggest that WR is a more complex process in which internal signals may cooperate with an individual’s susceptibility to gain weight, such as personality traits or cognitive style. Although emotions are integral to personality and the subject’s food choices, they may be guided by internal factors, which affect hunger and satiety. Psychological factors, including emotions, affect both body weight and weight management. Although some personality traits may change over one’s lifetime as they correlate with age [63], some of them undeniably strongly affect food choices and can predict future WR in each period of life. In summary, while personality traits, cognitive style, and emotions can help guarantee a stable weight and prevent WR, it is also necessary to attend to developed personal strategies, a healthy lifestyle (broadly understood), and a focus on goals.

The psychological predictors of weight regain that play the greatest role in body weight fluctuations, along with their mechanisms, are summarized in Table 1.

### 3.2. Neurobiological Factors in WR

#### 3.2.1. Food Addiction, Reward Sensitivity, and Neural Responses to Food Stimuli

Failure of long-term weight control can also be attributed to an obesogenic environment and the hedonic properties of food, both of which are associated with the neurobiological factors underlying weight regain. The hedonic properties of food exceed the need to maintain the body’s energy homeostasis, resulting in an increase in body weight and, consequently, obesity [64]. Among the major neurobiological systems involved in mediating reward aspects of food consumption are the dopaminergic, opioidergic, and cannabinoid systems. An obesogenic environment shapes energy homeostasis by affecting the cortical and limbic areas of the brain where the hedonic reward pathways are located, which promotes obesity and makes it difficult to maintain normal body weight [65]. The factors involved in this process include easy access to highly processed food that is characterized by a high energy density and promoted by marketing, encouraging unhealthy eating patterns; this affects an individual’s energy homeostasis. Food stimulates the brain centers responsible for pleasure and reward, which translates into an increased willingness to consume tasty food even when energy requirements have already been met [66].

Functional magnetic resonance imaging (fMRI) studies have shown that unique food experiences are generated in the orbitofrontal cortex (OFC), which transmits dopaminergic inputs via the mesocortical pathway [67,68]. The OFC is involved in decision-making, integrating sensory information about taste, smell, and visuals, and may play an important role in excessive eating behaviors [69]. This region is strongly associated with addictive and obsessive–compulsive disorders, and it responds to food stimuli, such as flavor and texture; hyperactivating its response to food cues may increase the risk of weight gain. fMRI methods are widely used in research into the brain activation associated with the development of obesity. These methods detect stimuli-induced activation of neuronal activity and connectivity associated with changes in perfusion and hemoglobin oxygenation [70,71]. One of the most frequently used paradigms in fMRI research into obesity is cue reactivity, which evokes central reactions in response to some specific cues and tasks, such as visual and sensory food stimuli. These methods may be helpful in understanding which areas of patients’ brains show altered activity and can predict future perspectives for weight loss and WR. Similarly to drug users, individuals with obesity experience activation of specific areas of the brain associated with reward anticipation and habit learning in response to food stimuli, regardless of physiological states of hunger and satiety [72]. The brain reward mesocorticolimbic circuitry is responsible for the reward response to consumption. Central structures are implicated in this circuitry, including the ventral tegmental area (VTA), the nucleus accumbens (NAc), the prefrontal cortex (PFC), and the amygdala; these, in combination with neurotransmitters, play an important role in appetitive motivation [73,74,75]. The amygdala, parahippocampal gyrus, and cingulate gyrus are particularly engaged in the emotional control of food intake [76]. These structures, which are mainly composed of dopaminergic neurons, are also involved in addictive behaviors, decision-making, and emotional processing [71]. Similar activity in these brain regions is observed during drug cue presentation to individuals with substance use disorder, where activation of the ventral striatum (VS), OFC, PFC, ACC, insula, and amygdala is induced [77].

Obesity is associated with significant changes in brain structure, connectivity, and functional activity. In comparison with lean individuals, people with obesity show greater activation in the brain regions involved in reward and flavor processing, particularly in response to high-calorie food odor stimuli [78]. Moreover, differences in the reward and attention networks are not only observed during food-related tasks, but also in resting-state conditions [79,80,81]. Individuals with obesity demonstrate a heightened attentional bias toward food cues and increased activation in parietal and visual cortices, which may promote excessive food consumption and weight gain [82,83]. Enhanced reactivity to highly palatable food cues is consistently found in key reward-related brain areas, such as the insula, amygdala, striatum, anterior cingulate cortex (ACC), and orbitofrontal cortex (OFC), even in a state of satiety [84,85,86,87]. These findings suggest that food intake in obesity is driven primarily by nonhomeostatic mechanisms. Additionally, increased activation has been observed in regions implicated in executive functioning, memory, emotional regulation, and self-referential processing [88]. Conversely, individuals with obesity exhibit decreased activation of the left dorsolateral prefrontal cortex (DLPFC) following meal consumption [89,90], indicating impaired self-regulatory control. Structural brain alterations have also been documented. Obesity is linked with reduced gray matter (GM) and white matter (WM) volume and density, particularly in areas responsible for executive function and inhibitory control [91,92,93,94,95]. Decreased GM volume in the precentral gyrus (PCG) and putamen has been associated with impaired self-control and dysregulated sensorimotor processing, both contributing to overeating behaviors [92]. Furthermore, obesity is characterized by deficits in planning, decision-making, and problem-solving, closely related to diminished prefrontal cortex (PFC) functioning [96]. These neurobiological alterations, interacting with psychological factors, play a pivotal role in shaping the individual traits and predispositions that affect WR. Notably, key features such as the ability to self-control appear to be partially determined by underlying neurobiological pathways. The relationship between neurobiology, behavioral outcomes, and WR is summarized in Table 2.

People with obesity and individuals with substance use disorders show impairments in the pathways of the dopaminergic system which are associated with the sensitivity-to-reward trait, as well as with self-control, interoceptive awareness, stress reactivity, and conditioning [97]. Sensitivity to reward is associated with the mesolimbic dopamine system and related to an increased risk of addiction. Higher body weight is positively correlated with an increased sensitivity-to-reward trait, along with higher anhedonia levels [35]. These results suggest that long-term overeating contributes to a reduction in the sensitivity of the reward system, leading to a compensatory mechanism for excessive food intake due to the reduced amount of dopamine D2 receptors (D2Rs) in the striatum, increased expression of dopamine transporters, and decreased activity of the dopamine pathway [98,99,100]. Similarly, in the course of chronic drug use, neuroplastic changes and the so-called tolerance can occur in the dopaminergic system. This leads to a reduction in D2Rs and dopamine in the striatum, a decrease in neuroadaptation, and consequently reduced sensitivity of the reward regions; it is then necessary to increase the dose to experience the same degree of reward. It has been speculated that people with obesity exhibit the same addiction pattern after exposure to tasty food stimuli, leading directly to excess caloric intake [101,102,103,104,105] (Figure 2). These findings support the “reward deficit theory”, which suggests that diminished dopaminergic signaling may lead to compensatory overeating as a means of achieving reward, thus contributing to both obesity development and WR. This mechanism resembles that observed in drug addiction, as both groups show similar reward circuit engagement. Additionally, previous meta-analyses and morphometric data have found that this mechanism is associated with GM hypodensity in the OFC, medial PFC, thalamus, and midbrain among patients with overweight or obesity, which suggests structural vulnerabilities that could predispose individuals to dysregulated eating and difficulties in long-term weight control [69,91,106]. Moreover, lower-density brain regions responsible for motivated behavior (which include the OFC, medial PFC, ACC, and amygdala) are associated with higher levels of impulsiveness [107,108,109,110], a personality trait that correlates positively with increased susceptibility to food addiction and emotional overeating, both of which are significant risk factors for WR. The study of Kerr et al. [111] noted that individuals with higher levels of impulsiveness exhibited increased activation in the ACC and bilateral amygdala during reward anticipation. It was observed that impulsiveness showed a positive correlation with neural activity in the ventral ACC and bilateral amygdala in response to reward cues. These results indicate that variations in the functioning of these brain regions contribute to differences in reward sensitivity which, in turn, shape key aspects of personality. Furthermore, studies have shown that food addiction is associated with impulsiveness, higher body mass, deficits in food-related inhibitory control, and severity of depressive symptoms, with lower values being associated with successful weight reduction [112,113]. Based on these data, it can be assumed that WR may be also due to mechanisms that depend on dopamine deficiency, particularly in combination with inadequate stress-coping strategies. Research has also indicated there to be some differences in regional brain glucose metabolism between healthy controls and participants with compulsive substance use disorders or binge eating disorders. These differences concern D2R availability and glucose metabolism in the OFC and anterior cingulate gyrus of the prefrontal areas, which were observed in both drug-addicted subjects and individuals with obesity [100,114]. The reduced availability of D2Rs in patients with obesity is associated with altered glucose metabolism in parietal regions associated with taste perception; this potentially heightens the drive for highly palatable food [115,116,117]. Another emerging hypothesis suggests that reduced numbers of D2Rs may decrease the motivation for physical activity, rather than solely increasing appetite, thus contributing to WR through decreased energy expenditure [118]. Taken together, the current evidence suggests that disruptions in the dopaminergic reward system—characterized by diminished D2 receptor availability, impaired reward sensitivity, structural brain changes, and heightened impulsivity—may collectively underpin vulnerability to WR.

Obesity is associated with abnormal functioning in homeostasis-dependent brain regions and in areas involved in hedonic pathways. In response to visual food stimuli, activation increases in several frontal regions (including the orbitofrontal cortex (OFC) and the anterior cingulate cortex (ACC)) and mesolimbic regions (including the caudate, putamen, nucleus accumbens (NAc), insula, ventral tegmental area (VTA), hippocampus (HIPP), and amygdala). Activation however decreases in the dorsolateral prefrontal cortex (DLPFC), which is engaged in inhibitory control [84,85,86,87].

#### 3.2.2. Metabolic Adaptations

Weight loss and changes in weight composition lead to homeostatic changes known as metabolic adaptations, which are widely described as biological drivers of WR. Metabolic adaptations involve persistent reductions in the resting metabolic rate (following adjustments for changes in body composition), aimed at increasing cues for energy intake, decreasing energy expenditure, and improving metabolic efficiency [119]. Daily energy requirements, along with total energy expenditure, fall to around 15% below the values that would be expected on the basis of post-loss body weight [120].

The biological mechanisms responsible for food intake are localized in the hypothalamus, where appetite control is regulated. Neuropeptides, adipokines, and gastrointestinal hormones are implicated in the process of energy balance maintenance [121]. Regulation of food intake is associated with five hypothalamic nuclei, namely the arcuate nucleus (ARC), the paraventricular nucleus (PVN), the ventromedial nucleus (VMN), the lateral hypothalamic area (LHA), and the dorsomedial region (DMV). These regions maintain the body’s metabolic homeostasis through orexigenic and anorexigenic peptides, which affect appetite control [121]. Two antagonistic systems located in the ARC are composed of orexigenic and anorexigenic neurons. Orexigenic neurons, such as agouti-related peptide (AgRP) and neuropeptide Y (NPY), express hunger-stimulating substances that result in increased food intake and reduced energy expenditure under starvation conditions. Conversely, the anorexigenic system, composed of pro-opiomelanocortin (POMC) and cocaine- and amphetamine-regulated transcript (CART) neurons, is responsible for inhibiting caloric intake and increasing energy expenditure in a state of surplus. Peripheral signals of satiety and hunger are also generated from the digestive system and fat tissue, including anorexigenic hormones like leptin, insulin, cholecystokinin (CCK), peptide YY (PYY), glucagon-like peptide 1 (GLP-1), and orexigenic ghrelin. Gut hormones transfer satiety and hunger signals to the brain centers that control appetite and to centers within the central nervous system (CNS), particularly the hypothalamus and brainstem [122,123,124].

Weight loss interventions may affect the neurobiological mechanisms involved in appetite and satiety control and thus predispose to WR. Energy intake reduction begins with an anti-starvation mechanism, which is not a part of a homeostatic feedback system [125]. During energy restriction, changes in appetite hormones are observed, including in ghrelin, leptin, and PYY; these result in sustained appetite enhancement and increased caloric intake [126]. Weight loss induces a substantial decrease in the satiety hormone leptin, which regulates energy homeostasis by inhibiting hunger [127,128,129]. In addition, levels of the appetite-stimulating hormone ghrelin increase during weight loss [130,131]. These hormonal changes after weight loss, which result in satiety disorders and increased hunger, favor WR through elevated energy consumption, as has been previously documented [132,133].

Neuroimaging studies have documented neurobiological brain alterations associated with differential reactions to food which may be used to predict WR. These are related to changes in plasma gut hormone concentrations. A systematic review by Zanchi et al. [134] showed that orexigenic and anorexigenic hormones differently affect the same brain regions engaged in food consumption, with higher ghrelin plasma levels being associated with increased activity of the amygdala, insula, and PFC and with hypoactivity in subcortical regions (including thehypothalamus), while plasma levels of leptin, insulin, glucose, GLP-1, CCK, and PYY affect these same areas in the opposite manner [134]. Ghrelin-mediated neural responses to food images increase in the amygdala, anterior insula, and OFC, which encode food values. This indicates that activation of neural circuitry in the human brain in the regulation of appetite can be modulated by gut peptides, consequently affecting eating behavior. Hormonal alterations are also observed after BS. However, these changes depend on the surgical procedure [135,136].

The hormones involved in hunger and satiety control and the changes seen after weight loss interventions are described in Table 3.

### 3.3. The Effects of Dietary, Pharmacological, and Bariatric Approaches on WR

Weight loss interventions, including dietary strategies, pharmacotherapy, and bariatric surgery, have been shown to induce significant changes not only in food choices but also in brain structure, function, and neurocognitive processes. These modifications may have an effect on the risk of WR and long-term weight maintenance success. Understanding the distinct and overlapping effects of the various therapeutic approaches on neural mechanisms is essential if obesity treatment outcomes are to be optimized.

#### 3.3.1. Dietary Weight Loss Interventions

Dietary interventions are the most common weight loss approaches and may alter food choices. Opstal et al. [171] demonstrated that an eight-week hypocaloric diet decreased neural activity in regions associated with reward processing and feeding behavior—including the insula, OFC, and frontal gyri—suggesting potential normalization of hedonic food-related brain circuits. Similarly, behavioral weight loss programs led to proportional reductions in GM volume in the left PFC [172]. Successful dieters (≥7% weight loss) exhibited additional decreases in GM volume in the left insula, putamen, and PCG—regions linked to self-regulatory and sensorimotor functions. Moreover, higher GM volume in the orbitofrontal and parahippocampal cortices and greater WM in the orbitofrontal and temporal regions correlated with greater weight loss, emphasizing the relevance of brain areas involved in motivation, taste processing, and proprioception [172]. Haltia et al. [173] has shown that WM volume in individuals with obesity was greater than in lean individuals, but that this relationship reversed after the diet intervention. However, they did not observe that dieting affected GM.

Cognitive control of food intake, also referred to as dietary restraint, appears to be critical to successful weight maintenance. Positron emission tomography (PET) imaging studies have revealed that increased activation in cognitive control regions is positively associated with higher dietary restraint among long-term weight maintainers from the National Weight Control Registry (NWCR) [174]. In response to meal consumption, participants who successfully adhered to dieting exhibited increased activation of the DLPFC, dorsal striatum, and anterior cerebellar lobe, as well as reduced activation of the OFC, as compared to non-dieters. Moreover, non-dieters showed higher activation of the OFC than successful dieters, and in both groups, OFC activation was inversely correlated with DLPFC activation [174]. NWCR maintainers were previously shown to be more persistent in specific behavioral strategies, such as frequent self-monitoring of body weight and food intake, adherence to a low-fat diet, and engagement in regular physical activity. Collectively, these behaviors contributed to a 29% weight reduction, which was successfully maintained over the next 5.5 years [175]. These findings highlight the strong association between activation of the DLPFC and dietary (as well as behavioral) restraint, as this brain region is implicated in the conscious regulation of emotions and emotion inhibition. Additionally, the higher OFC activation observed in unsuccessful individuals may be associated with overeating and compulsive food consumption, making it more difficult to achieve stable body weight. Su et al. [176] further found that unsuccessful dieters exhibited heightened activity in regions related to habit learning, addictive behaviors, and compulsive eating (such as the inferior OFC, caudate, and HIPP), while successful dieters showed increased activation in executive control regions like the middle frontal gyrus and cerebellum [176].

Importantly, Cornier et al. [177] reported differences in brain activity and neuronal responses to food stimuli among normal-weight individuals, individuals with obesity, and those who had lost weight. Their findings suggest that people with obesity and those who have lost weight exhibit similar abnormal neuronal responses to food, which play a crucial role in shaping eating behavior. After two days on a eucaloric diet, in a fasted state, lean individuals exhibited greater activation of the insula and inferior visual cortex in response to images of highly palatable foods compared to obese individuals who had lost weight. However, after two days of overeating, this response was attenuated in normal-weight individuals, whereas individuals with obesity who had lost weight continued to show persistent activation of the brain regions responsible for food motivation, even in a state of overfeeding [177]. Individuals who had lost weight exhibited greater activation of the insula, which is a brain region engaged in eating behaviors and rewards in response to taste, compared to lean individuals [178]. The insula reacts to smell, flavor, and swallowing in response to hunger and visual food stimuli [68,179,180], and as mentioned before, smaller GM volume in the insula is observed in successful dieters. Notably, hyperactivity or dysfunctions within the insula have been linked to disordered eating patterns, including anorexia nervosa (AN) [181,182,183]. Despite their contrasting clinical presentations, both obesity and AN share abnormalities in food-related neural circuits. Whether heightened insular activity predisposes to WR or emerges as a compensatory adaptation following weight loss requires further investigation.

Furthermore, a six-month fMRI prospective study, which investigated the association between early weight loss intervention effects on striatal food cue reactivity and BMI measures among participants, has shown that early brain process alterations might predict weight loss intervention outcomes [85]. The authors of the study reported that diet intervention and weight loss affected food cue reactivity in several striatum regions, such as the bilateral putamen, right pallidum, and left caudate. Given that the putamen is involved in reward processing in response to food cues, it has been suggested that changes in neural processes within the putamen might be used in future studies as a marker of intervention efficacy.

However, restrictive low-calorie diets and strict calorie monitoring may induce chronic stress [184]. Higher levels of chronic stress are linked to increased connectivity between the putamen and amygdala, as well as strengthened connectivity between the amygdala, ACC, and PFC in response to high-calorie food cues [185]. The amygdala regulates appetite in response to emotions and is activated by food cues. It is composed of multiple nuclei, such as the central nucleus of the amygdala (CeA), the basolateral nucleus of the amygdala (BLA), and the medial nucleus of the amygdala (MeA), which can induce food intake even in a satiety state [186], and its reaction to food cues is related to circulating levels of the orexigenic ghrelin [187]. The NPY peptide, expressed in the CeA and BLA, is involved in emotional eating, as it has both an anti-anxiety effect and one of the strongest known feeding-inducing properties [188,189]. NPY expression in CeA is elevated in response to stress, particularly in combination with a high-calorie diet, leading to insulin resistance in the central amygdala [189]. Accelerated brain activation in striatal–limbic regions responsible for emotion processing and reward—such as the amygdala—has been correlated with higher leptin levels in individuals with obesity [190]. These results align with studies demonstrating a connection between obesity, emotional regulation difficulties, and heightened stress levels [19,191]. Importantly, stress-induced neuroplasticity in the BLA and MeA leads to dendritic hypertrophy and synaptic remodeling, producing long-term consequences [192,193]. These findings suggest that chronic exposure to stress is associated with an increased brain response to food and may predispose to an increased risk of obesity via unhealthy eating habits. Given that the amygdala is closely associated with stress reactions and responses to negative stimuli, diet-induced stress may predispose to WR.

#### 3.3.2. Bariatric Surgery

Reduced self-control, reduced executive function, and reduced inhibitory control are associated with a lack of weight control. One solution to these problems is surgical treatments of obesity, such as BS, which is effective not only because it reduces the volume of the stomach or the route through which food passes, but also because it leads to changes in brain activity, which can result in a different response to food stimuli. The most widely used procedures in clinical practice are RYGB and LSG, both of which result in a similar, but not identical, long-term weight loss effect [194]. It has been documented that the connectivity of the OFC to the limbic regions in obese individuals decreases after BS, yielding an improvement in controlling eating behaviors [195]. A recent meta-analysis showed that improvements in executive control, memory, and attention are observed among BS patients following surgery [196]. People with obesity are shown to have decreased activation in the regions engaged in cognitive control or responsible for inhibition in response to visual food cues, which is why this mechanism of BS may be especially useful for those struggling with failures of weight loss interventions over their lifespan as a result of the hedonic processes associated with the reward system. Moreover, studies have shown that pre-surgery patients with lower GM and WM density compared to controls see an increase in both densities following BS [94,197,198,199]. Finally, RYGB has been shown to restore neuronal plasticity to normal six months after surgery among patients with obesity [200]. Additionally, an increase in the visual cortex was positively correlated with heightened levels of circulating GLP-1 following weight loss [200].

BS seems to allow for the recovery of brain obesity-related structures engaged in food motivation, improving greater GM and WM integrity and increasing cognitive functioning. However, structural brain restoration may not be sufficient for all patients undergoing BS. Although bariatric surgery typically results in substantial weight loss (30–45%), approximately 10–20% of patients experience significant WR, often defined as recovering 10–25% of the initially lost or excess weight [201,202,203]. Weight maintenance after BS is influenced by a variety of psychosocial factors that may contribute to WR. Significant WR predictors after BS include poorer mental health, depressive symptom occurrence, low socioeconomic status [204,205], and a tendency toward addictive behavior [206], although the main factor that predisposes to WR is broadly understood to be eating disorders, occurring both pre- and postoperatively [204,205,206,207,208]. Marshall et al. [56] state that patients should be assessed prior to surgery for dichotomous thinking. They demonstrated that weight loss after BS among patients was completely mediated by dichotomous thinking about food. Psychological support may improve BS outcomes and prevent future WR associated with rigid beliefs about diet, which are an integral part of eating disorders.

#### 3.3.3. Pharmacotherapy

Recent years have seen a growing body of evidence regarding the use of anti-obesity medications (AOMs) in the treatment of obesity. Current guidelines indicate that obesity pharmacotherapy should be considered for patients with BMIs of ≥30 kg/m^2^ and for those with BMIs ≥ 27 kg/m^2^ and at least one weight-related comorbidity [6]. Anti-obesity drugs are especially advised as adjunctive therapy alongside lifestyle intervention in the cases of patients who are unable to lose and sustain weight and who struggle with chronic obesity [209]. AOMs approved by the US Food and Drug Administration include orlistat, phentermine-topiramate, naltrexone-bupropion, liraglutide, semaglutide, tirzepatide, and setmelanotide (this last one is for the treatment of genetic obesity only) [210]. The mechanism of action of these drugs includes a reduced appetite, decreased energy absorption, or the promotion of energy expenditure [211]. The effectiveness of AOMs includes not only reducing body weight but also a modest improvement in cardio-metabolic profile [212]. Moreover, in the setting of WR, AOMs have also been shown to successfully prevent weight gain after both diet-induced and bariatric surgery-induced weight loss, as demonstrated in several studies (Table 4) [213,214,215,216].

In a recent retrospective observational study, Cifuentes et al. [215] demonstrated that the use of AOMs following very-low-calorie diets utilizing a total meal replacement program significantly attenuated WR compared to no pharmacological intervention. After 18 months, patients using AOMs regained an average of 3.29 kg (31.5% of weight loss), while those without AOMs regained 7.61 kg (52.16% of weight loss). These differences were statistically significant (*p* = 0.006 for weight in kg; *p* = 0.04 for percentage). The retrospective study by Gazda et al. [217] evaluated the effectiveness of various treatment strategies for post-bariatric WR and found that after 3 months patients on GLP-1-RA lost 4.5% of their body weight, compared to 2.2% in the non-GLP-1-RA group and 1.4% in the lifestyle modification group (*p* < 0.001). The authors concluded that GLP-1-RA-based therapies were the most effective in treating WR, regardless of the type of bariatric surgery. Also, in study by Jensen et al. [218] among patients experiencing WR after bariatric surgery, 67.4% of WR was safely lost with liraglutide and semaglutide treatment. The study by Stanford et al. [219] examined the utility of AOMs in patients with inadequate weight loss or WR after RYGB and SG and showed that patients treated with topiramate were twice as likely to lose at least 10% of total weight compared to those on other AOMs or those who received only lifestyle interventions (odds ratio = 1.9, *p* = 0.018). Furthermore, a study by Toth et al. [220] showed that the use of AOMs (especially topiramate and phentermine) at the weight plateau phase can help young adults after RYGB maintain further weight loss. In this context, a study by Istfan et al. [221] confirmed that phentermine and topiramate are particularly effective in reducing WR after RYGB, providing hope for more effective management of this issue. Similarly, a preventive effect of AOMs on post-bariatric WR was demonstrated in the study by Srivastava and Buffington [222] and Stanford et al. [223]. In conclusion, these studies indicate that AOMs are highly effective in mitigating WR and can be an essential adjunct to BS, particularly in patients who experience weight plateau or inadequate weight loss after the initial post-surgical period. The use of medications like topiramate, phentermine, or GLP-RAs can significantly enhance weight loss outcomes, whether initiated early or at the point of WR.

**Table 4 nutrients-17-01662-t004:** FDA-approved AOMs—mechanisms of action and selected data on their impact on weight regain.

AOMs	Weight-Lowering Mechanisms of Action	Influence of WR
Orlistat	Inhibits lipase—blocks fat absorption in the intestine	after diet-induced weight loss: Richelsen et al. (2007) [224]: Following initial very-low-energy diet-induced weight loss (14.4 ± 2.0 kg), orlistat treatment (120 mg/3 times a day over 3 years) resulted in lower weight regain (4.6 ± 8.6 kg vs. 7.0 ± 7.1 kg; *p* < 0.02) and a higher proportion of patients achieving ≥5% weight loss compared to placebo. Sjöström et al. (1998) [225]: In a 2-year randomized controlled trial, orlistat (120 mg/3 times a day) resulted in greater weight loss compared to placebo (10.2% vs. 6.1%; *p* < 0.001) during the first year on a slightly hypocaloric diet (600 kcal/day deficit) and less weight regain in the second year (0.9 kg loss vs. 2.5 kg regain; *p* < 0.001). after BS-induced weight loss: Zoss I et al. (2002) [226]: Orlistat (120 mg/3 times a day for 8 months) resulted in greater weight loss compared to controls (8 ± 3 kg vs. 3 ± 2 kg; *p* < 0.01) in patients with adjustable gastric banding who had stopped losing weight.
Phentermine-topiramate	Phentermine suppresses appetite via norepinephrine; topiramate affects satiety through CNS effects	after BS-induced weight loss: Istfan, N.W. et al. (2020) [221]: WR after RYGB in patients treated with phentermine/topiramate, topiramate, or phentermine was approximately 10% lower at the end of the 6-year observation period compared to placebo. Schwartz, J. et al. (2016) [227]: Phentermine and topiramate effectively reduced weight loss (12–13%) in patients who experienced WR or a weight loss plateau after RYBG/LAGB.
Naltrexone-bipropion	Bupropion modulates dopamine/norepinephrine; naltrexone blocks food-related reward pathways	no data
Liraglutide [193]	GLP-1 receptor agonist—increases satiety, delays gastric emptying, and reduces appetite	after BS-induced weight loss: de Moraes et al. (2024) [228]: A systematic review and meta-analysis (16 studies, 881 individuals, mean follow-up time from 3 months to 4 years) showed that liraglutide led to significant reductions in BMI (−8.56 kg/m^2^; *p* < 0.01) and a mean reduction in total weight (−16.03 kg; *p* = 0.05) in patients who experienced WR after BS. Vinciguerra et al. (2024) [229]: Meta-analysis (119 individuals) showed that liraglutide at 3 mg led to weight loss (5.6 ± 2.6% at 12 weeks and 9.3 ± 3.6% at 24 weeks) with a significant reduction in waist circumference (*p* < 0.0001) in patients who experienced inadequate weight loss or WR after BS. Jamal et al. (2024) [230]: In patients who underwent SG, 3-month liraglutide treatment resulted in a significant mean weight loss of 5.94 kg (6.2% of pre-treatment weight; *p* < 0.001), with greater weight reduction observed in patients aged 31–40 years and those tolerating doses ≥2.4 mg, suggesting liraglutide as an effective and dose-responsive adjunct therapy for managing WR or inadequate weight loss post-surgery.
Semaglutide	GLP-1 receptor agonist—increases satiety, delays gastric emptying, and reduces appetite	after BS-induced weight loss: Lautenbach, A. et al. (2022) [231]: In patients who experienced WR or inadequate weight loss after BS, semaglutide treatment led to a mean total weight loss of 6.0% at 3 months and 10.3% at 6 months. Murvelashvili, N. et al. (2023) [232]: In patients with post-bariatric surgery WR, semaglutide (1.0 mg weekly) led to superior weight loss (−12.92%) compared with liraglutide (−8.77%). Kanai, R. et al. (2024) [233]: In patients with obesity and T2D after LSG, semaglutide treatment (1.0 mg weekly) resulted in additional BMI reductions (−1.6 kg/m^2^) and improved glycemic control.
Tirzepatide	Combination of glucose-dependent insulinotropic polypeptide (GIP) and GLP-1 receptor agonist synergistically improves appetite control and insulin sensitivity	Stoll, F. et al. (2025) [234]: In post-bariatric (SG, RYGB) patients who experienced WR or insufficient weight loss (12.0 ± 3.4%; *p* < 0.001), tirzepatide treatment led to significant weight loss and improvements in metabolic health, regardless of surgery type or sex.

In the context of WR, it is important to emphasize that cessation of anti-obesity pharmacotherapy is independently associated with subsequent weight gain. During the randomized controlled trial by Sjöström et al. [225], patients who switched from orlistat to placebo experienced a mean weight regain of 2.5 kg over one year. Wilding et al. [235] reported that one year after discontinuation of semaglutide treatment, individuals regained approximately two-thirds of the weight they had initially lost. Similarly, significant WR was also demonstrated one year after stopping liraglutide [236] or tirzepatide [237] therapy.

Importantly, WR may also occur despite ongoing pharmacologic treatment. The study by Weintraub et al. [238] aimed to assess long-term weight loss outcomes with FDA-approved and off-label AOMs, revealing that over a 2.5- to 5.5-year period patients regained an average of 6.5% of their initial body weight. The underlying mechanisms and the magnitude of this challenge remain poorly understood and highlight the need for further long-term, high-quality research.

The potential of AOMs to prevent WR or stimulate weight loss in patients experiencing inadequate weight loss may be related to several mechanisms, including eating disorders. GLP-1 RAs influence eating behavior by acting on the midbrain’s VTA and reduce highly palatable food consumption by suppressing dopamine neuron signaling [239]. Activation of GLP-1 receptors has been shown to decrease motivation for food reward by interacting with the mesolimbic system in rats [240]. The hunger-suppressing action of GLP-1 RAs is thus utilized in obesity treatment, as they affect feeding behavior and induce satiety by modulating POMC neurons [241,242]. On the other hand, emotional eating has been linked to decreased sensitivity to the central effects of GLP-1 receptor activation. Emotional eaters exhibit an increased brain response to food cues in regions involved in reward processing and satiety, such as the insula, amygdala, and OFC, but may be less responsive to satiety-regulating signals like GLP-1 [243]. Moreover, participants struggling with emotional eating show reduced sensitivity to the inhibitory effects of GLP-1 RAs in regions such as the left amygdala, right OFC, bilateral insula, and left caudate nucleus in response to visual food cues [244]. These findings suggest that not all patients may benefit from GLP-1 RAs in terms of weight loss, especially those whose obesity and WR are strongly linked to emotional factors, which may reduce sensitivity to physiological satiety signals.

A promising pharmacotherapy for long-term weight loss management is a combination of two drugs with different mechanisms of action, i.e., naltrexone/bupropion (NB). These drugs work synergistically by increasing activity in the melanocortin system of the hypothalamus [245]. NB interacts with the mesolimbic reward system, decreasing appetite and also increasing the activity of brain areas responsible for self-control and awareness by reducing hypothalamic reactivity to food cues [246]. NB treatment is associated with decreased functional connectivity density in the superior parietal cortex and medial PFC and increased connectivity with regions involved in salience attribution or interoception, such as ACC and insula [246]. Additionally, NB pharmacotherapy, in combination with lifestyle interventions, may be successfully used in individuals with obesity and binge eating by minimizing emotional eating and lowering binge eating episodes and other pathological eating behaviors [247,248,249,250]. NB therapy is a promising treatment in weight loss and weight management programs, especially in patients with emotion-related eating patterns, and it may prevent future WR.

## 4. Discussion

This review has highlighted the interplay between selected psychological traits, emotional regulation, and neurobiological factors in weight regain (WR) following weight loss. Our findings suggest that WR is a multifactorial phenomenon shaped by overlapping psychological and biological processes, rather than isolated determinants [11,14] (Figure 3). Psychological factors such as impulsiveness, dichotomous thinking, and low self-control appear to interact with alterations in brain regions involved in reward processing and cognitive control, collectively driving vulnerability to WR [20,24,28,30,107,108,109,172,174].

Although previous studies have separately addressed psychological and neurobiological predictors, this review is the first, to our knowledge, to synthesize both perspectives. By integrating these domains, we provide a more comprehensive framework for understanding WR while underlining the dynamic relationship between emotional–behavioral tendencies and brain function. This approach addresses an important gap in the existing literature, which has largely treated psychological and neurobiological factors as distinct.

Clinically, our findings suggest that early psychological assessments—particularly screening for impulsiveness and rigid cognitive patterns—could enhance risk stratification for WR. Interventions aimed at improving emotional regulation skills and strengthening cognitive control mechanisms may improve long-term outcomes. Importantly, early detection of psychological symptoms that predispose to future relapse might allow for the implementation of preventive strategies before WR occurs, breaking the vicious cycle of repeated weight loss and regain. Routine psychological screening in patients with overweight and obesity, prior to initiating weight loss interventions, should be considered a key step in personalized treatment planning. Although integrating neuroimaging biomarkers into clinical practice remains challenging due to economic and logistic limitations, future advances in this field may eventually support individualized treatment planning.

Several limitations must be acknowledged. First, as a narrative review, our study may be subject to selection bias and does not offer meta-analytic quantification. Second, the psychological and neurobiological factors discussed are not exhaustive: we have focused on the selected determinants most supported by the literature. Third, most of the studies we have included were cross-sectional or based on small samples, which limits the ability to infer causality. Future research should therefore adopt longitudinal designs, explore broader psychological constructs (such as motivation and resilience), and further investigate how surgical and pharmacological interventions can modify the psychological and neurobiological pathways associated with WR.

In conclusion, WR is best understood as a dynamic, biopsychological phenomenon. Targeting both psychological vulnerabilities and neurobiological dysregulations offers a promising direction for the development of more effective, individualized strategies for long-term weight maintenance.

Excessive body weight motivates an individual to reduce body weight. However, maintenance of reduced weight is affected by many external factors (including environment, lifestyle, and socioeconomic status), as well as by internal factors. Internal factors are those related to psychological aspects (including character traits, ways of dealing with emotions, ways of thinking) and neurobiological aspects (including metabolic adaptations, brain responses to food cues, and adipocyte inflammation). All of these factors can align to favor weight regain. Weight gain can then prompt the person to attempt to lose weight again, leading to a vicious circle.

## 5. Conclusions and Future Perspectives

Psychological and neurobiological factors are closely interconnected and together affect the regulation of body weight, making long-term weight maintenance particularly challenging. In the treatment of obesity, where the goal is sustained weight loss, psychological support appears indispensable. The development of emotional awareness and effective emotional regulation strategies should be key components of interventions for individuals prone to body weight fluctuations across their lifespan, as emotional balance is one of the most critical predictors of long-term weight control. Weight regain should be considered, at least in part, a consequence of impaired emotional regulation mechanisms, closely linked to brain function.

Future research should further explore the role of psychotherapy in the management of WR with the goal of deepening our understanding of the bidirectional relationship between eating behaviors and the functioning of specific brain structures.

## Figures and Tables

**Figure 1 nutrients-17-01662-f001:**
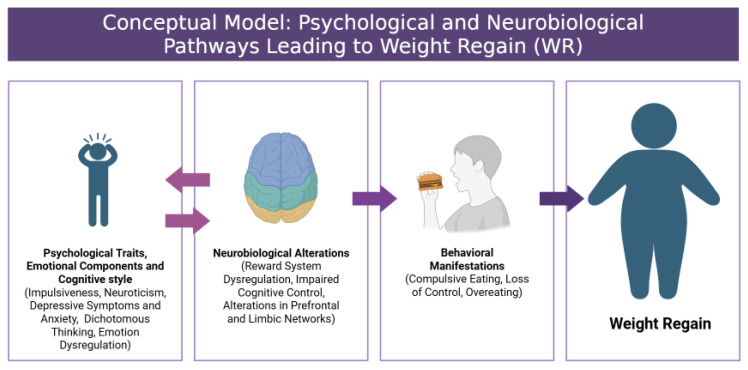
Conceptual model: psychological and neurobiological pathways leading to weight regain.

**Figure 2 nutrients-17-01662-f002:**
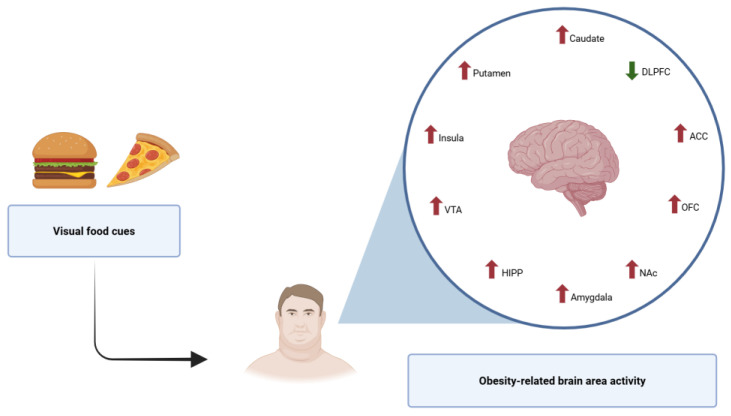
Brain area activation in response to food cues in obesity (figure created using BioRender.com). 
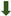
 decrease activity 
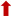
 increase activity.

**Figure 3 nutrients-17-01662-f003:**
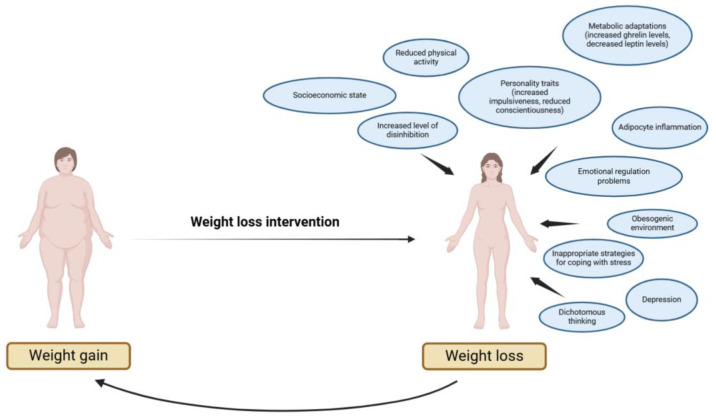
Summary of factors predisposing to weight regain after weight loss intervention (figure created using BioRender.com).

**Table 1 nutrients-17-01662-t001:** A summary of the predictors of psychological weight regain most responsible for weight cycling, along with their mechanisms.

Psychological Factor	Effect on Weight Regain	Mechanism of Weight Control
Impulsiveness	↑ WR risk	Increased impulsiveness leads to heightened activation in the ACC and bilateral amygdala during reward anticipation, promoting overeating. Hypodensity of OFC, ACC, amygdala, and medial PFC is also correlated with higher levels of impulsiveness.
Neuroticism	↑ WR risk	Higher neuroticism correlates with emotional eating and increased susceptibility to weight regain due to heightened amygdala activity and reduced connectivity between the amygdala and ACC, which may favor anxiety and depressive symptoms.
Dichotomous thinking	↑ WR risk	Dichotomous thinking contributes to unhealthy eating patterns, overeating, and weight cycling through highly rigid “black and white” dietary restraint. This cognitive style also mediates the relationship between depression and obesity and may be closely associated with perfectionism.
High sensitivity to reward	↑ WR risk	Sensitivity to reward is associated with impairments in dopaminergic system pathways and may favor an increased risk of food addiction. At higher levels, it promotes compensatory overeating in response to negative stimuli, as this feature is related to emotional regulation problems.
Conscientiousness	↓ WR risk	Conscientiousness promotes sustained efforts in maintaining weight, linked to stronger prefrontal cortex activity supporting self-regulation.
High self-control	↓ WR risk	Higher levels of self-control predict successful weight maintenance, which is associated with better regulation of eating behavior and reduced activation of reward areas.
Persistence	↓ WR risk	It prevents weight regain and enables long-term weight maintenance through consistent use of weight management strategies.
High self-efficacy	↓ WR risk	It fosters persistence in pursuing defined goals and makes it easier to engage in behaviors that promote a healthy lifestyle. In combination with higher self-control, it can increase motivation.

Abbreviations: ↑ WR risk: increased risk of weight regain; ↓ WR risk: decreased risk of weight regain; ACC: anterior cingulate cortex; OFC: orbitofrontal cortex; PFC: prefrontal cortex.

**Table 2 nutrients-17-01662-t002:** A summary of the relationship between brain regions, their behavioral outcomes, and the risk of WR.

Brain Region	Behavioral Outcomes	Impact on Weight Regain (WR)
Orbitofrontal cortex (OFC)	Hedonic evaluation of food; reward valuation; decision-making	Hyperactivity leads to excessive food reward seeking and overeating; lower density is associated with higher levels of impulsiveness
Ventral striatum (VS)	Reward anticipation and habit learning	Increased activation enhances susceptibility to food cues
Prefrontal cortex (PFC)	Executive control; inhibition of impulsive behaviors	Reduced activity impairs self-regulation, facilitating overeating; lower density is associated with higher levels of impulsiveness
Anterior cingulate cortex (ACC)	Cognitive and emotional processing; motivation and decision-making; learning and cost–benefit analysis	Impairments associated with higher impulsivity and food addiction
Amygdala	Emotional regulation (processing of emotions, particularly fear and anxiety); emotional eating	Hyperresponsivity enhances emotional eating and craving; lower density is associated with higher levels of impulsiveness; increased activity is positively correlated with neuroticism
Ventral tegmental area (VTA)	Reward processing; motivation and goal-directed behavior; learning and memory; addictive behaviors	Dysregulation reduces reward sensitivity, promoting compensatory overeating
Nucleus accumbens (NAc)	Incentive salience; reinforcement learning; hedonic value	Overactivation reinforces habitual overeating patterns
Parahippocampal gyrus	Emotional memory; emotional processing; contextual modulation of food cues	Altered activation in response to high-calorie food cues may enhance context-driven food craving and emotional eating; dysfunctional activity is associated with impaired inhibitory control over eating behaviors
Insula	Interoceptive awareness; sensory integration (taste and odor perception); cognitive control and decision-making	Altered interoception linked to dysregulated eating; heightened insular activation in response to food cues is associated with stronger cravings, greater emotional reactivity, and reduced control over eating behavior
Thalamus and midbrain	Sensory relay; reward processing integration	Structural deficits are associated with impaired motivated behavior

**Table 3 nutrients-17-01662-t003:** Hormones involved in hunger and satiety control and changes in them after weight loss interventions.

Hormone	Mechanism of Action	Serum Concentrations in Obesity	Changes After Bariatric Surgery	Changes After Calorie Reduction Diet
Ghrelin	↑ appetite	↓	↑ LAGB [137,138,139,140,141,142] ↓ LSG [143,144,145,146,147] RYGB: ↓ in short-term/↑ in long-term [148]	↑ [130,132,149,150]
Leptin	↓ appetite improves satiation	↑ (leptin resistance)	↓ LAGB [138,140,148,151] ↓ LSG [143,145,146,152] ↓ RYGB [138,145,148,152,153]	↓ [132,154,155]
Insulin	↓ appetite	↑ (insulin resistance)	↓ LAGB [138,140] ↓ LSG [145,146,152] ↓ RYGB [138,145,152]	↓ [132,150,154,156,157]
CCK	slows gastric emptying induces satiation	postprandial ↓	? LAGB ↑ LSG [158,159] ↑ RYGB [159,160]	↓ [132,161]
GLP-1	↓ appetite slows gastric emptying	postprandial ↓	- LAGB [148,153] ↑ LSG [145,147,162] ↑ RYGB [145,148]	↓ [132,157,163]
PYY	↓ appetite slows gastric emptying	postprandial ↓	↑/- LAGB [164,165,166,167,168] ↑ LSG [145,147,162] ↑ RYGB [145,148]	↓ [132,169,170]

Abbreviations: LAGB: laparoscopic adjustable gastric banding; LSG: laparoscopic sleeve gastrectomy; RYGB: Roux-en-Y gastric bypass; CCK: cholecystokinin; GLP-1: glucagon-like peptide 1; PYY: YY peptide; ↑: increase; ↓: decrease; (-): no change; (?): no data.

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
