# Peer review of "The Interplay Between Psychological and Neurobiological Predictors of Weight Regain: A Narrative Review"

_nutrients, 2025, doi:10.3390/nu17101662_

Round 1

Reviewer 1 Report (Previous Reviewer 3)

Comments and Suggestions for Authors

After going over the previous version of the paper, and comparing the current revised version. The author/s already made the appropriate revisions.

Author Response

We are grateful to you for taking the time to read our paper and for your constructive comments which help us improve our manuscript.

Reviewer 2 Report (Previous Reviewer 4)

Comments and Suggestions for Authors

This paper lack in form and content e needs relevant changes, as indicated below:

First of all what is main aim and rational of the study, why the scientific literature in need to this paper. Moreover I may expect that a narrative review to be conducted by an expert in the field of obesity treatments, otherwise I may feel that alternatively authors should conduct a systematic review.

In this direction authors should justify their choice for writing a narrative review and not a systematic one.  Moreover going through this review, it is still far from the standards of the narrative review guideline. Please check: https://cdn.amegroups.cn/journals/tgh/files/journals/28/articles/6919/public/6919-PB2-5118-R1.pdf

In particular:

  • The title is generic
  • The abstract is poorly presented, in sub paragraphs, never seen before
  • The keywords are a repetition to those already mentioned in the title, and this is not the case, where in general those should be different to increase the chance to find the article by a broader search strategy
  • The introduction section wrongly include a part that should be under the method section
  • The method section is not available with all it important parts: search strategies, keyword, MeSH terms, inclusion and exclusion criteria etc.
  • The results of the review should be reported under a defined section, and the findings should be summarised a table
  • The discussion section is broad, speculative and naïf. It needs to be rewritten properly including clearly the following subsections:
  • The main findings of the review
  • The clinical implication of the review
  • Strengths and limitations  
  • The new directions for the future research 

Author Response

We are grateful to you for taking the time to read our paper and for your constructive comments. We have carefully reviewed the comments and have revised the manuscript accordingly. Our responses are given below in a point-by-point manner. All suggested corrections have been included in the modified version and highlighted in yellow in the text.

We hope the revised version is now suitable for publication, and we look forward to hearing from you due course.

Point-by-point response to Comments and Suggestions for Authors:

Comments 1: First of all what is main aim and rational of the study, why the scientific literature in need to this paper. Moreover I may expect that a narrative review to be conducted by an expert in the field of obesity treatments, otherwise I may feel that alternatively authors should conduct a systematic review.

In this direction authors should justify their choice for writing a narrative review and not a systematic one.  Moreover going through this review, it is still far from the standards of the narrative review guideline. Please check: https://cdn.amegroups.cn/journals/tgh/files/journals/28/articles/6919/public/6919-PB2-5118-R1.pdf”

Response 1: Thank you for this comment. After carefully reviewing your suggestions, we have rewritten the manuscript in accordance to the Narrative Review checklist. In particular:

1) In the Abstract - we have included sections: introduction, objective, methods, results and conclusion. Sentences regarding Future perspective and Study limitations have also been added.

2) The Introduction has been shortened, removing content on the treatment and epidemiology of obesity. A brief justification of the rationale/need for this work has been added (see page 2, lines 24-28). This section also includes the Conceptual Model recommended by another reviewer.3) The Methods section containing information necessary for narrative review such as search strategy, search MeSH terms, years considered, language, inclusion and exclusion criteria has been added (see page 3).The SANRA checklist to assess quality of this narrative review have been also introduced.4) The Discussion section have been rewrite to highlight the key findings, study limitations and future perspectives of the study.

Comments 2: “The title is generic”

Response 2: Thank you for your suggestion. The title has been revised on “Interplay between psychological and neurobiological predictors of weight regain: A Narrative Review”. (previously: “A vicious circle of obesity: psychological and neurobiological predictors of weight regain. A narrative review”.)

Comments 3: „The abstract is poorly presented, in sub paragraphs, never seen before”

Response 3: Thank you for the suggestion. The abstract has been revised into a single paragraph containing structured elements such as Introduction, Objectives, Methods, Results, and Conclusions.The abstract also incorporates a comment from another reviewer, who recommended mentioning the limitations of the review or gaps in the literature.

Comments 4:„The keywords are a repetition to those already mentioned in the title, and this is not the case, where in general those should be different to increase the chance to find the article by a broader search strategy”

Response 4: The keywords have been revised from: “obesity; weight regain; weight management; sensitivity to reward; motivation; self-control; psychology; neurobiology; neuroimaging; eating behavior” into: “obesity; weight control; reward-related eating; hedonic hunger; personality, psychological traits; brain function; motivation; self-control; eating behavior”

Comments 5: „The introduction section wrongly include a part that should be under the method section”

Response 5: Thank you very much for pointing out this aspect. The informations regarding the methods has been removed from the Introduction section, and section “2. Methods” has been introduced (see page 3)

Comments 6: „The method section is not available with all it important parts: search strategies, keyword, MeSH terms, inclusion and exclusion criteria etc.”

Response 6: Thank you for the suggestion. The section Methods with information about search strategy, search terms, inclusion and exclusion criteria, and assessment of studies analyzed in this narrative review (SANRA checklist) was added to the manuscript(see page 3, and Supplementary Material).

Comments 7:

„The results of the review should be reported under a defined section, and the findings should be summarised a table”

Response 8: Thank you for your comment. We revised the results section and added a tables under each subsection to summarize findings.

Page 8: Table 1. Summary of the most important psychological weight regain predictors and their mechanisms responsible for weight-cycling.

Page 10: Table 2. Summary of the relationship between brain regions, their behavioral outcomes, and risk of WR.

Page 19: Table 4. FDA-Approved AOMs – Mechanisms of Action and Selected Data on Their Impact on Weight Regain.

Comments 9:

„The discussion section is broad, speculative and naïf. It needs to be rewritten properly including clearly the following subsections:

  • The main findings of the review
  • The clinical implication of the review
  • Strengths and limitations  
  • The new directions for the future research „

Response 9:Thank you for pointing this out.We have thoroughly revised theDiscussion section to improve its clarity, structure, and scientific rigor. The section has been reorganized to explicitly include the following subsections: (1) main findings of the review, (2) clinical implications, (3) strengths and limitations, and (4) future research directions (please see page 23). We trust that the revised version more clearly communicates the key messages and addresses the Reviewer’s concerns.

 4. Response to Comments on the Quality of English Language

Point 1: The English is fine and does not require any improvement.

Response 1: Since another reviewer suggested a language edit, the entire manuscript was subjected to English language proofreading by native speaker.

5. Additional clarifications

We would like to admit that considering all the Reviewer’s suggestions, our manuscript underwent extensive revisions and was generally improved.

Reviewer 3 Report (New Reviewer)

Comments and Suggestions for Authors

General Comments
1.    The paper addresses an important and under-explored aspect of obesity management—weight regain (WR) with a commendable interdisciplinary approach combining psychology and neurobiology.
2.    The narrative style allows flexibility in covering a broad scope, and the review is well-referenced.
3.    The manuscript is excessively long and, at times, redundant.
4.    The MS lacks critical synthesis and hierarchical organization of reviewed evidence.
5.    The methodology for the literature search is vague and does not meet rigorous review standards.
6.    The review sometimes drifts into explanatory summaries without clear linkage to WR prediction or intervention.
Comments by section
Abstract
1.    The abstract is overly detailed; it can be shortened to focus more on key findings and conclusions.
2.    There is no mention of limitations of the review or literature gaps.
Introduction
1.    The writing is wordy, so I suggest condensing some epidemiological and treatment contexts to focus more quickly on the WR phenomenon.
2.    A conceptual model (e.g., psychological traits → brain pathways → behaviour → WR) can guide the reader early in the framing.
Methods
1.    The section lacks transparency and rigour.
2.    There are no PRISMA-style flowcharts, inclusion/exclusion criteria, or quality assessment of studies reviewed.
3.    The search strategy is simplistic; Boolean search terms are listed but poorly integrated.
Psychological Factors of WR
Personality Traits
1.    The discussion leans heavily on correlational findings; causality is often assumed without caveats.
2.    Too many individual studies are presented without integration or synthesis.
3.    Repetitive references to studies throughout paragraphs. 
Emotional Components
1.    A stronger discussion on how emotion dysregulation differs by gender, life stage, or comorbidities is needed. 
2.    The section lacks nuance on bidirectional influences between mood and WR.
Dichotomous Thinking
1.    The section needs clarification. Is dichotomous thinking a trait, a coping mechanism, or a cognitive style?
2.    More integration with intervention literature would be beneficial.
Neurobiological Factors of WR
Reward System & Food Addiction
1.    Many neuroimaging claims are made with weak integration of meta-analytic evidence.
2.    The discussion in this section would benefit from a table summarizing brain regions, behavioural outcomes, and WR risk.
3.    At times, the section conflates obesity neurobiology broadly with WR-specific mechanisms.
Metabolic Adaptations
1.    The section is dense, technical, and hard to follow without clearer subheadings or summaries.
2.    It requires a more apparent distinction between adaptation in general and those predictive of WR specifically.
Brain Structure and Activity
1.    It is sometimes repetitive with Section 3.1; consider merging or reorganizing.
2.    Using figures is helpful but should be accompanied by clear captions and explanations of how they were derived (original or adapted?).
Bariatric Surgery
1.    The section lacks a nuanced discussion of individual differences in BS outcomes.
2.    The authors should reference psychosocial predictors of success after BS. 
Pharmacological Management
1.    The role of emotionally driven eating in pharmacotherapy responsiveness is briefly mentioned but underexplored.
2.    A table summarizing drugs, mechanisms, and WR effects would aid clarity.
Discussion
1.    This section repeats earlier content rather than synthesizing or critically evaluating evidence.
2.    The authors should highlight research gaps, practical implications, and future directions.
3.    The authors overstate certain conclusions without meta-analytic support (e.g., “Dichotomous thinking may be the strongest predictor”).
Language and Style
1.    The manuscript is overly long, with many redundant or loosely structured paragraphs.
2.    Editing for conciseness, active voice, and terminological precision is highly recommended.
3.    I suggest using more summary tables and conceptual models to reduce textual repetition.

Comments on the Quality of English Language

English can be slightly improved

Author Response

We are grateful to you for taking the time to read our paper and for your constructive comments. We have carefully reviewed the comments and have revised the manuscript accordingly. Our responses are given below in a point-by-point manner. All suggested corrections have been included in the modified version and highlighted in yellow in the text.

We hope the revised version is now suitable for publication, and we look forward to hearing from you due course.

Point-by-point response to Comments and Suggestions:

Comments 1: “The paper addresses an important and under-explored aspect of obesity management—weight regain (WR) with a commendable interdisciplinary approach combining psychology and neurobiology.”

Response 1: Thank you for recognizing the significance and potential of the topic discussed in this narrative review.

Comments 2:“The narrative style allows flexibility in covering a broad scope, and the review is well-referenced.”

Response 2:Thank you for highlighting the strengths of the narrative approach and the quality of referencing in our review

Comments 3: The manuscript is excessively long and, at times, redundant.

Response 3: We agree with this suggestion.The MS have been generally improved and and the duplicated information have been removed. In the revised version, we have carefully reviewed the content to eliminate unnecessary repetition and combined overlapping sections where appropriate to improve conciseness and cohesion. However, based on other reviewer suggestions—including the expansion of the Introduction, Methods, and Discussion sections, and the inclusion of additional tables for clarity—the overall length has remained relatively unchanged. We have made every effort to streamline the manuscript while ensuring scientific rigor and completeness.

Comments 4:The MS lacks critical synthesis and hierarchical organization of reviewed evidence.

Response 4:We appreciate the reviewer’s constructive feedback. In response, we have revised the manuscript to improve the critical synthesis of the reviewed literature, emphasizing relationships between psychological and neurobiological factors. Additionally, the structure has been reorganized to follow a clearer hierarchical framework. These changes aim to improve clarity, logical flow, and coherence of argumentation throughout the manuscript.

Comments 5:  The methodology for the literature search is vague and does not meet rigorous review standards.

Response 5:Thank you for the suggestion. The section Methods with information about search strategy, search terms, inclusion and exclusion criteria, and assessment of studies analyzed in this narrative review (SANRA checklist) have been introduced to the manuscript (see section 2. Methods, page 3and Supplementary Material).

Comments 6:The review sometimes drifts into explanatory summaries without clear linkage to WR prediction or intervention.

Response 6:We have carefully revised the manuscript to ensure that each explanatory summary is clearly connected to the context of weight regain, particularly in terms of its predictive value or relevance to therapeutic intervention. We clarified the rationale behind including specific neurobiological and psychological mechanisms and emphasized their direct or indirect contributions to the development or prevention of weight regain, thereby improving coherence and maintaining a focused narrative throughout the review.

Comments 7:

“Abstract
1.    The abstract is overly detailed; it can be shortened to focus more on key findings and conclusions.
2.    There is no mention of limitations of the review or literature gaps.”

Response 7:

The abstract have been shortened and key results and conclusions have been highlighted.

Sentences regarding future perspective and study limitations have also been added.

Comments 8:

“Introduction
1.    The writing is wordy, so I suggest condensing some epidemiological and treatment contexts to focus more quickly on the WR phenomenon.
2.    A conceptual model (e.g., psychological traits → brain pathways → behaviour → WR) can guide the reader early in the framing.”

Response 8:

Thank you for your valuable suggestions.

1.The introductory section has been revised and condensed to reduce wordiness, particularly in the parts related to epidemiological and treatment context. This allowed for a quicker transition to the core topic of the manuscript.

  1. To better illustrate the complex interplay between psychological and neurobiological factors contributing to weight regain, we developed a conceptual model (Figure 1, see Introduction section).

Comments 9:

“Methods
1.    The section lacks transparency and rigour.
2.    There are no PRISMA-style flowcharts, inclusion/exclusion criteria, or quality assessment of studies reviewed.
3.    The search strategy is simplistic; Boolean search terms are listed but poorly integrated.”

Response 9:Thank you for your valuable insight. In order to make the methodology more transparent the Methods section containing information necessary for narrative review such as search strategy, search MeSH terms, years and language considered, inclusion and exclusion criteria has been added (see page 3) .The SANRA checklist to assess quality of this narrative review have been also introduced. Because our study is narrative review, not systematic-review the PRISMA-style flowcharts have not been introduced. For the same reason, and due to the fact that the objectives of this narrative review is interdisciplinary, less rigorous literature search criteria were used.

Justifications for choosing narrative review instead of systematic review were added in the Abstract and Introduction section (see page 3, lines 1-4).

Comments 10:

Personality Traits
1.    The discussion leans heavily on correlational findings; causality is often assumed without caveats.
2.    Too many individual studies are presented without integration or synthesis.
3.    Repetitive references to studies throughout paragraphs. 

Response 10: Thank you for these important observations. The section has been revised to ensure that correlational findings are presented with appropriate caveats and without implying causality. Additionally, several individual study descriptions were consolidated to improve clarity and promote synthesis of the reviewed evidence. Repetitive citations have been reduced, and the narrative was refined to highlight overarching patterns and relationships rather than isolated results.

Comments 11:

Emotional Components
1.    A stronger discussion on how emotion dysregulation differs by gender, life stage, or comorbidities is needed. 
2.    The section lacks nuance on bidirectional influences between mood and WR.

Response 11:

  1. A discussion on gender and age-related emotion dysregulation has been added (see page 6, lines: 5-27).
  2. The bidirectional relationship has been added as the summary of all the data in the ‘Emotional Component’ subsection (see page 6, lines: 28-36).

Comments 12:

Dichotomous Thinking
1.    The section needs clarification. Is dichotomous thinking a trait, a coping mechanism, or a cognitive style?
2.    More integration with intervention literature would be beneficial.

Response 12:

Thank you for your suggestions. We have revised the manuscript to define dichotomous thinking more preciselyand in general we have expanded this whole section to include psychotherapies and their clinical implications for WR.

Comments 13:
Reward System & Food Addiction
1.    Many neuroimaging claims are made with weak integration of meta-analytic evidence.
2.    The discussion in this section would benefit from a table summarizing brain regions, behavioural outcomes, and WR risk.
3.    At times, the section conflates obesity neurobiology broadly with WR-specific mechanisms.

Response 13: Thank you for your comment. We generally revised the manuscript and we connected “Brain Structure and Activity” along with “Reward System and Food Addiction” section  in subsection titled “Food Addiction, Reward Sensitivity, and Neural Responses to Food Stimuli” (see page 9) to make a clearer context of this topic. Moreover, we provided Table 2 titled “Summary of the relationship between brain regions, their behavioral outcomes, and risk of WR”. We hope that the revisions satisfactorily address the Reviewer’s expectations.

Comments 14:

Metabolic Adaptations
1.    The section is dense, technical, and hard to follow without clearer subheadings or summaries.
2.    It requires a more apparent distinction between adaptation in general and those predictive of WR specifically.

Response 14: We thank the Reviewer for this valuable feedback. While we recognize that the “Metabolic Adaptations” section is complex due to the nature of the content, we believe it provides essential context for understanding the physiological basis of weight regain (WR). In response to the comment, we did not substantially alter the technical depth of the section, as we consider this information critical; however, we improved the readability by restructuring the content into shorter, more digestible paragraphs. This adjustment enhances clarity without compromising scientific detail. Additionally, to address the request for a clearer distinction between general metabolic adaptations and those specifically predictive of WR, we refined several transitions and expanded the discussion by linking metabolic changes to neuroimaging findings. These findings help illustrate how certain adaptations may contribute to altered brain activity patterns that predispose individuals to WR, thus better integrating the physiological and behavioral dimensions of the topic.We hope these changes sufficiently improve the section’s accessibility and address the Reviewer’s concerns.

Comments 15:

Brain Structure and Activity
1.    It is sometimes repetitive with Section 3.1; consider merging or reorganizing.
2.    Using figures is helpful but should be accompanied by clear captions and explanations of how they were derived (original or adapted?).

Response 15: As mentioned above in response 13, this section was connected to section “Reward System and Food Addiction” and was generally reorganized. We appreciate the Reviewer’s comment regarding the figures. All figures included in the manuscript are original and were created by the authors. Each caption has been carefully composed to provide a clear and informative description of the figure’s content and relevance. Additionally, we have specified the software tools used in the creation of the figures to ensure full transparency. We hope this clarification addresses the Reviewer’s concern.

Comments 16:

Bariatric Surgery
1.    The section lacks a nuanced discussion of individual differences in BS outcomes.
2.    The authors should reference psychosocial predictors of success after BS. 

Response 16: We thank the Reviewer for highlighting the importance of considering individual differences and psychosocial predictors in bariatric surgery outcomes. While the primary aim of our review was not to provide an exhaustive discussion of bariatric surgery, we recognize the relevance of these factors in the context of weight regain (WR). In response, we have expanded the relevant paragraph in the bariatric surgery section to incorporate a brief but focused discussion of individual variability in outcomes. Specifically, we now highlight psychosocial determinants such as depressive symptoms, socioeconomic status, addictive tendencies, and especially the presence of eating disorders both before and after surgery. Furthermore, we reference recent findings indicating that dichotomous thinking may fully mediate weight loss outcomes post-surgery, underscoring the potential utility of pre-surgical psychological assessment and support (see pages: 17-18).

Comments 17:

Pharmacological Management
1.    The role of emotionally driven eating in pharmacotherapy responsiveness is briefly mentioned but underexplored.
2.    A table summarizing drugs, mechanisms, and WR effects would aid clarity.

Response 17:We have substantially revised the Pharmacotherapy section in accordance with the Reviewer’s suggestions (see pages 18–22). Additionally, we have added a table titled “FDA-Approved AOMs – Mechanisms of Action and Selected Data on Their Impact on Weight Regain,” which aims to enhance clarity and provide a concise summary of the current evidence.

Comments 18:

Discussion
1.    This section repeats earlier content rather than synthesizing or critically evaluating evidence.
2.    The authors should highlight research gaps, practical implications, and future directions.
3.    The authors overstate certain conclusions without meta-analytic support (e.g., “Dichotomous thinking may be the strongest predictor”).

Response 18: Thank you for pointing this out. We have thoroughly revised the Discussion section to improve its clarity, structure, and scientific rigor. The section has been reorganized to explicitly include the following subsections: (1) main findings of the review, (2) clinical implications, (3) strengths and limitations, and (4) future research directions (please see page 22). We trust that the revised version more clearly communicates the key messages and addresses the Reviewer’s concerns. To highlight the WR phenomenon as a main topic of this narrative review and to justify the need of this paper sentences in the Introduction section have been added.

Comments 19:

Language and Style
1.    The manuscript is overly long, with many redundant or loosely structured paragraphs.
2.    Editing for conciseness, active voice, and terminological precision is highly recommended.
3.    I suggest using more summary tables and conceptual models to reduce textual repetition.

Response 19:

As suggested, the entire manuscript has been proofread in English (certificate attached).

Additional clarifications

We would like to admit that considering all the Reviewer’s suggestions, our manuscript underwent extensive revisions and was generally improved.

Round 2

Reviewer 2 Report (Previous Reviewer 4)

Comments and Suggestions for Authors

.

Reviewer 3 Report (New Reviewer)

Comments and Suggestions for Authors

This manuscript offers a valuable and timely contribution to the literature on obesity management by focusing on the psychological and neurobiological predictors of weight regain (WR). The interdisciplinary narrative approach is well-justified and enhances the paper’s clinical and theoretical relevance. I commend the authors for their thoughtful and thorough revisions, significantly improving the manuscript’s structure, clarity, and focus. Including a conceptual model, detailed tables, and improved methodological transparency (including the SANRA checklist) significantly strengthens the work. The expanded discussion on emotional regulation, dichotomous thinking, and integrating neural mechanisms with personality traits provides a coherent and nuanced synthesis. While the manuscript remains relatively long, the authors have made commendable efforts to reduce redundancy and improve readability without sacrificing scientific depth. Minor language and stylistic issues remain, but do not detract significantly from comprehension. Overall, the revised version meets the standards of a high-quality narrative review and is suitable for publication with only minor editorial polishing.

This manuscript is a resubmission of an earlier submission. The following is a list of the peer review reports and author responses from that submission.

Round 1

Reviewer 1 Report

Comments and Suggestions for Authors

Thank you for this interesting manuscript and fascinating concept development! This review is very well written and flows well. As a reviewer, I have no significant changes to recommend. However, I do have quite a few editorial changes to recommend. I have listed them by line number on each page below and for ease have included my recommendation for how the text should be revised to read... rather than stating the current wording and saying to change the text. I recognize that the page numbers when I printed the manuscript may not exactly match what you will print, however, I felt this would be the best I could do to identify the editorial changes. 

Abstract:

line 17: This narrative review

line 26:  ... studies have shown

line 28: ... still it cannot be stated

Page 2

line 2: ... obesity has increased significantly, including in the pediatric population

line 7: ... bariatric surgery (BS)

line 34: This paper aims

line 46: ... and weight control also include reward 

Page 3

line 9: Moreover, Buchanan et al. (18), using structural equation modeling (SEM), demonstrated the relationship between binge eating and decreased self-efficacy.

line12: ... and less likely to succeed...

line 36:... changes in BMI throughout adulthood.

line 45: ... revealed, the women who successfully..

Page 4

line 18: ...research has shown that in...

Page 5

line 26: ...suggest that WR is a complex process...

line 29: ... by internal factors, which influence hunger and...

Page 7

line 7: Furthermore, studies have shown...

Page 8

line 12: Weight loss induces a substantial decrease...

line 14: In addition, levels of ghrelin, which is a hormone that stimulates appetite, drop during weight loss. NOTE: I think this is what was meant but I was not sure!

line 17: ... energy consumption and were previously documented [96,97].

line 18: ... also observed after BS. However, these changes differ with each surgical procedure [98,99].

Table 1: Please spell out BS for the table heading because it is a heading and needs to stand alone.

Page 9.

line 26: ... to a meal among obese individuals...

Page 10:

In the last lines of the Figure 1 description, beginning with the word "figure" the words all run together.

line 25: after the words "feeding behavior" there seems to be some words missing as the next phrase " was following caloric restriction" does not seem to be in the correct place.

line 37: In turn, Haltia et al [156] has shown that...

Page 11

line 17: ... for the next 5.5 years [158].

line 25: In turn, a group of successful dieters exhibited a higher activation...

Page 12

line 13: An important role in appetitive motivation is also played by the amygdala...

line 29: Accelerated bran activation in striatal-limbic regions are responsible...

line 44: Bariatric surgery (BS), brain and WR

line 49: ... food, but also through changes...

Page 13

line 9: ...may be especially useful for those individuals...

line 16: ...levels of circulating GLP-1 following weight loss

line19: ... may prevent future relapse

line 31: .. or promotion of energy expenditure

line 39: ... emptying and as a result..

line 42: ... weight loss. A RCT study, The SCALE Maintenance, assessed...

line 47:... in the placebo group. Moreover, 

Page 14

line 11: ... and the left caudate..

line 45:... more frequent binge eating episode occurrences and...

line 50: ... addictive behaviors and may predispose a person to.. 

Page 15

line 22: ... for better understanding of neurobiological factors...

line 23: It is important to determine if a hypocaloric diet... 

Thank you very much for including the abbreviations at the end of the manuscript, Please reorder the abbreviations into alphabetical order so a specific one is easier for the reader to find. Also, please doublecheck that the abbreviation is spelled out in the text when it is first introduced. 

NOTE As a reviewer, I did not check the references as to their accuracy. However, I did notice that in some of the references the words ran together so please doublecheck the spacing in all of the references in your reference list.

Thank you again for a fine manuscript.

Reviewer 2 Report

Comments and Suggestions for Authors

“Who is to blame for weight regain: psyche or brain? A narrative Review”( nutrients-3526118)

This manuscript aimed to provide a comprehensive overview of the mechanisms of weight regain by a narrative review. The authors draw conclusions that it still cannot be drawn unequivocally if the psyche or brain (e.g., neuroendocrine control of eating) has a greater effect on WR. This conclusions can be achieved even without the current manuscript. Overall, this topic is important and interesting. However, some concerns appeared after reading the whole manuscript.

  1. After reading the title, I assume that the authors would firstly provide the psychological and neurobiological factors contributing to WR, respectively. And then some comparison might be done to draw some conclusions about which factors might be more important. However, it is not the case after reading the whole manuscript.

  1. The organization of the current manuscript is a little confusing since only one part was about the psyche part while the remaining parts are about the brain part, which makes the readers might lead to the conclusions that this manuscript is mainly focused on brain part while the answer in the title should be that neurobiological factors would be more important than psychological ones. Moreover, the psychological factors are far from comprehensive. “personality”and “emotion”can not summarize all the psychological factors involved.

  1. The latest data about obesity should be updated.

Global, regional, and national prevalence of adult overweight and obesity, 1990–2021, with forecasts to 2050: a forecasting study for the Global Burden of Disease Study 2021. Lancet, March 03, 2025

Global, regional, and national prevalence of child and adolescent overweight and obesity, 1990–2021, with forecasts to 2050: a forecasting study for the Global Burden of Disease Study 2021. Lancet, March 03, 2025

  1. I am not quite sure about the novelties of the current review since some related reviews have been published while neglected in the current manuscript.

Busetto, L., Bettini, S., Makaronidis, J., Roberts, C. A., Halford, J. C., & Batterham, R. L. (2021). Mechanisms of weight regain. European journal of internal medicine, 93, 3-7.

  1. It would be better to provide a figure to depict all the factors involved in weight regain to help the readers get an overview about this topic.

Reviewer 3 Report

Comments and Suggestions for Authors

here are some comments and suggestions/questions

- abstract - seems long, check if comply with journal word limit -should note the scope of the review, year, database search... how many resulting paper... abstract should be concise and clear and comply with the format of the journal

introduction is there any statistics for global obesity ... 

at the end of the introduction should note the scope of the review, at the least the year ranges of the studies reviewed

there should also be some logic in the presentation of the succeeding sections (or an overarching framework)

- initial though that come to mind, might be cognitive, affective, physiological or biological....

- there should also be some linkages between the different sections

limitations? what now? future perspective?

Reviewer 4 Report

Comments and Suggestions for Authors

1)    The title should be revised to appear more scientific

2)    The abstract should be structured in one paragraph divided in subsection as the journal’s guideline

3)    The keywords should differ from those that appear in the title or repeated in the abstract in order to increase the probability of finding the paper with a wider search strategy. 

4)    The introduction is poorly referenced especially since dealing with a review on a very popular topic

5)    A Method section should exist even if dealing with a narrative review. Please check for example and submit also the checklist: https://legacyfileshare.elsevier.com/promis_misc/ANDJ%20Narrative%20Review%20Checklist.pdf

6)    From section 2 to 7 should be included under one section as Results

7)    Why do authors summarize only the hormonal factor in a table and ignore the others? All results regarding all factors involved in weight regain need to be reported equally in a table.  

8)    A discussion section before the conclusion should be added to be structured as follow:

  • The main findings of the review
  • The clinical implications of this review
  • The strengths and limitations of this review, foremost authors should clearly explain why they did not conduct a systematic review on the topic
  • The new directions for future research on the topic

9)    Table 1 and Figure 1 are really poor and elementary, and need to be much elaborative.

10) The cognitive part has been completely ignored and needs to be reported and discussed, as well as the impact of the environment. Please check: https://www.tandfonline.com/doi/pdf/10.2147/dmso.s89836